# When recalling in-context, Transformers are not SSMs

## Abstract

Despite the advantageous subquadratic complexity of modern recurrent deep learning models – such as state-space models (SSMs) – recent studies have highlighted their potential shortcomings compared to transformers on reasoning and memorization tasks. In this paper, we dive deeper into one of such benchmarks: associative recall (AR), which has been shown to correlate well with language modeling performance, and inspect in detail the effects of scaling and optimization issues in recently proposed token mixing strategies. We first demonstrate that, unlike standard transformers, the choice of learning rate plays a critical role in the performance of modern recurrent models: an issue that can severely affect reported performance in previous works and suggests further research is needed to stabilize training. Next, we show that recurrent and attention-based models exhibit contrasting benefits when scaling in width as opposed to depth, with attention being notably unable to solve AR when limited to a single layer. We then further inspect 1-layer transformers, revealing that despite their poor performance, their training dynamics surprisingly resemble the formation of induction heads, a phenomenon previously observed only in their 2-layer counterparts. Finally, through architectural ablations, we study how components affects Transformer and Mamba's performance and optimization stability.

## 1 Introduction

Since early developments [Rumelhart et al., 1986, Elman, 1990], RNNs have driven progress in machine learning techniques for sequential data, with milestones such as Echo-State Networks [Jaeger, 2001] the LSTM [Hochreiter and Schmidhuber, 1997] and the GRU [Cho et al., 2014]. However, two problems severely limit the application of RNNs in modern times: first, GPU architectures designed for large matrix multiplications struggle with sequential processing. Secondly, it is widely known that recurrent models are hard to train due to vanishing and exploding gradients issues [Bengio et al., 1994, Hochreiter et al., 2001, Pascanu et al., 2013].

**Attention.** These challenges have led to the introduction of a different paradigm: the Attention mechanism, implemented around the Transformer architecture [Vaswani et al., 2017]. Instead of processing inputs sequentially while building up internal memory (RNNs), Attention computes large matrices of pairwise interactions between data points, allowing for modeling direct links between elements in a sequence and thus attenuating the vanishing gradient issue. While Attention, being based on matrix multiplications, is extremely GPU efficient, computing pairwise interactions results in $O(L^2)$ inference and memory complexity, where $L$ denotes the input sequence length. For this reason, techniques such as patching [Dosovitskiy et al., 2021, Pagnoni et al., 2024], gradient checkpointing [Chen et al., 2016], and FlashAttention [Dao et al., 2022, Dao, 2023, Shah et al., 2024] become of paramount importance when training and deploying Attention-based models at scale. Despite this limitation, Transformers successfully powers most state-of-the-art architectures we use today: beyond large language models [Devlin, 2018, Brown et al., 2020, Team et al., 2024],

Submitted to 39th Conference on Neural Information Processing Systems (NeurIPS 2025). Do not distribute.

Attention found widespread application in vision [Dosovitskiy et al., 2021, Touvron et al., 2021, Bertasius et al., 2021, Liu et al., 2024a], graph processing [Ma et al., 2023], and genome analysis domains [Dalla-Torre et al., 2024], among others.

Nevertheless, the quadratic complexity of Attention has remained a pressing limitation, prompting numerous efforts to develop more efficient approximations [Wang et al., 2020, Choromanski et al., 2020, Chen et al., 2021, Lee-Thorp et al., 2022]. Many of these approaches have even revealed connections to recurrent formulations [Katharopoulos et al., 2020, Schlag et al., 2021].

**SSMs and other linear token mixers.** More recently, we have witnessed a resurgence of RNNs in state-of-the-art industry-size applications such as language modeling. Sparked by the S4 model [Gu et al., 2020, 2022], which surpassed Attention-based models on long-range reasoning tasks [Tay et al., 2020], we have rapidly seen in the last year a drastic increase in the usage of RNNs in deep architectures, albeit in a linear[1] form that guarantees both $O(L)$ memory/inference complexity and fast computation on GPUs [Martin and Cundy, 2018, Orvieto et al., 2023] while matching or surpassing transformers on downstream tasks: a prime example are State-space Models (SSMs) such as Mamba(2) [Gu and Dao, 2024, Dao and Gu, 2024], along with variants based on similar ambitions [De et al., 2024, Peng et al., 2024, Yang et al., 2024a]. These novel fast recurrent processing strategies sparked the interest of many practitioners in the field, leading to novel applications in several domains, including vision [Liu et al., 2024b, Liang et al., 2024], audio generation [Goel et al., 2022], online learning [Zucchet et al., 2023] and reinforcement learning [Lu et al., 2023].

**Different expressivity?** It has been shown [Dao and Gu, 2024, Ali et al., 2024, Sieber et al., 2024], that one can put in direct correspondence Attention with SSM processing: due to the linearity of SSMs in the hidden state – the main distinction between SSMs and classical nonlinear RNNs [Cirone et al., 2024] – it is possible to write down the "attention matrix" corresponding to SSM processing at a given input. Yet, such a matrix is highly structured, a feature that boosts speeds at very high context lengths [Waleffe et al., 2024] but may hurt optimization due to iterated products involving the model parameters [Zucchet and Orvieto, 2024]. On top of this, the recurrent formulation of SSMs clearly points to potential memory issues compared to attention on simple yet important tasks such as copying [Jelassi et al., 2024] and associative recall [Arora et al., 2023].

**.. or perhaps just harder optimization?** Despite empirical results and worst-case bounds [Arora et al., 2024] regarding the capabilities of SSM models on simple yet important tasks, it is yet unclear if optimization issues, such as the curse of memory or vanishing gradients [Pascanu et al., 2013, Zucchet and Orvieto, 2024], confound or understanding of capabilities of new recurrent models. We found that this is indeed the case in associative recall [Arora et al., 2023], as we show in Figure 1.

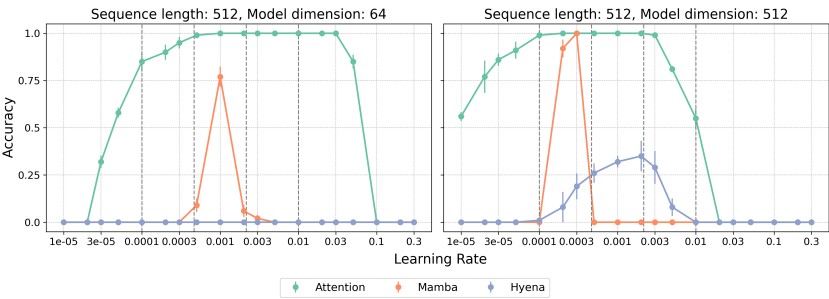

Figure 1: *We show the performance of Attention, Hyena and Mamba using an extensive learning rate grid search. Differently from attention, the window of suitable learning rates for Mamba and Hyena is relatively narrow. We also compare our grid search with the one used in Zoology [Arora et al., 2023] (**dashed vertical lines**) to highlight how the suitable learning rate can be missed. The results show the mean and relative max-min errors after 3 runs with different seeds.*

Figure 1 points to a crucial confounder when comparing SSM and attention capabilities: while fundamental expressivity issues exist between such model classes, the main driver of poor performance can be unsuccessful optimization. In a way, when considering basic prototypical yet challenging tasks, ***transformers are not SSMs* mainly because of their optimization dynamics.**

---

[1]Modern RNNs such as State-space Models are linear in the hidden-to-hidden state interactions, but have recurrent formulation that is non-linear in the input, see [Cirone et al., 2024]

In response to this issue, a potential confounder for future evaluations of new recurrent models, in this paper we take a closer look at associative recall and reveal new insights into the fundamental distinctions between attention models and recurrent models.

- **The Impact of Optimization on Recurrent Models**: We find that learning rate selection plays a pivotal role in the performance of recurrent models. More than just influencing performance, it determines whether these models can successfully solve the task. Overlooking this factor can lead to incorrect conclusion about their capabilities, highlighting the need for careful tuning when working with recurrent architectures.
- **Different Scaling Behavior of Depth and Width**: Our experiments reveal that these models differ in how they benefit from scaling width and depth. Consistent with prior research, recurrent models gain the most from increased width, as their reliance on hidden state updates makes a larger one beneficial for retaining information. Conversely, we show that a single-layer attention model fails to solve the task, while a two-layer version succeeds.
- **Training Dynamics and Induction-Like Phenomena in 1-layer architectures**: Expanding on the previous insight, we examine the behavior of single-layer models during training. Interestingly, even in this setting, attention exhibits a phenomenon reminiscent of induction heads—previously observed only in deeper models. We observe a drop in loss that does not correspond to improved performance, indicating that the model may struggle to leverage induction heads at this scale. Meanwhile, recurrent models show smoother training dynamics, and Mamba, in particular, demonstrates a steep performance increase similar to induction heads, even in a single-layer setup.
- **Ablation to reduce the gap between Transformers and SSMs**: Given the different behavior of the two architectures, we make a series of architectural changes to make the Transformer more similar to Mamba and viceversa. Our findings suggest that Mamba's advantages stem from more than just its convolutional, gating or architectural module, as its performance remains strong even when these components are removed or modified. We also give suggestions on how other architectures, can improve the baseline of Mamba in MQAR and its stability.

## 2   Background and Related works

**Associative Recall.**    With the rise of foundation models, deep learning has made significant advances, sparking growing interest in evaluating their reasoning capabilities. One key aspect of reasoning is the ability to recall previously encountered information. Intuitively, given the input

> "***Hakuna Matata*** *means **no worries** for the rest of your days.*
> *"**Hakuna Matata** means ...*"

a well-performing model should predict ***"no worries"*** with high likelihood. Building on this idea, the synthetic associative recall (AR) task, introduced by [Olsson et al., 2022], gained popularity as an efficient reasoning benchmark to assess promising model architectures at a relatively low cost. The task is structured as follows: Given a fixed Vocabulary $V$, each sample consists of a sequence of tokens sampled from $V$ representing alternating key-value pairs. Given such a sequence and a key that appeared earlier, the model must correctly infer its corresponding value: For example, given the input sequence:

$$A \ 6 \ I \ 9 \ C \ 7 \ P \ 1 \ S \ 4 \ D \ 2$$

and given the key $C \rightarrow \ ?$ the model should predict 7.

A crucial aspect of this task is that the tokens serve interchangeably as keys and values among samples—they are drawn from the same vocabulary rather than separate sets. Consequently, the model cannot rely on preassigned roles for tokens. Moreover, since token roles and positions vary across data points, the model cannot memorize a fixed mapping but must instead infer the correct associations dynamically in-context.

**Multi-Query Associative Recall.**    Building on previous research [Arora et al., 2023], our experiments employ a variation of AR known as multi-query associative recall (MQAR). This choice is motivated by the fact that standard AR is typically used to evaluate the ability of recurrent models to capture long-range dependencies using extremely long sequences—an area where attention-based models often struggle due to memory constraints. However, at the scale of our experiments, MQAR presents a more challenging and relevant task even with relatively small sequences.

There are two key distinctions between MQAR and its standard counterpart, both of which align more closely with the characteristics of natural language. First, it introduces a significantly larger vocabulary: from the 50 tokens of standard AR to approximately $8,000$ tokens in MQAR. This makes the task more representative of real-world language processing where the vocabulary size is in the order of hundreds of thousands of words. Second, instead of recalling a single key-value pair, MQAR requires the model to retrieve multiple values based on multiple queries. This more accurately mirrors the nature of language, where meaning is often derived from groups of words and interrelated concepts rather than isolated tokens. For instance, given an input sequence

$$A \ 6 \ I \ 9 \ C \ 7 \ P \ 1 \ S \ 4 \ D \ 2$$

and given multiple keys

$$C \to \ ? \ \ A \to \ ? \ \ D \to \ ?$$

we ask the model to recall the relative values 7, 6 and 2. Notably, if we were to restrict the model to retrieving only one key-value pair, the task would reduce to AR. We opted for this variant because prior studies have demonstrated that it more effectively highlights the differences between attention-based and recurrent models. By incorporating these linguistic properties, multi-query associative recall serves as a more insightful benchmark for evaluating model performance. Even if all of our analysis are made using multi-query associative recall, throughout this work we will use the terms AR and MQAR interchangeably for simplification.

**Induction heads.** While investigating the capabilities of transformers in few-shot learning, previous work ( [Olsson et al., 2022]) showed the phenomenon of induction heads. The main insight from this work was that during training, with transformers with at least 2 layers, a special kind of attention heads called "induction heads" is formed, causing a noticeable drop in the loss perplexity, while giving a sudden boost in In-context learning performances.

More formally, induction heads are implemented by a circuit consisting of a pair of attention heads in different layers that work together to copy or complete patterns. The first attention head copies information from the previous token into each other tokens, making it possible for the second attention head to attend to tokens based on what happened before them, rather than their own content. Specifically, the second head (the proper "induction head") searches for a previous place in the sequence where the present token **A** occurred and attends to the next token (call it **B** ), copying it and causing the model to be more likely to output **B** as the next token. That is, the two heads working together cause the sequence ...[**A**][**B**]...[**A**] to be more likely completed with [**B**].

Induction heads are named by analogy to inductive reasoning, where we might infer that if **A** is followed by **B** earlier in the context, **A** is more likely to be followed by **B** again later in the same context. Induction heads are capable of crystallizing that inference. They search the context for previous instances of the present token, attend to the token which would come next in the pattern repeated, and increase its probability in terms of logit. Induction heads attend to tokens that would be predicted by basic induction (over the context, rather than over the training data).

**Transformers and SSMs.** Let $X \in \mathbb{R}^{N \times d}$ a generic input consisting of $N$ elements in $d$ dimensions. Basic state-space models (SSMs) [Gu and Dao, 2024] compute outputs via a recurrence:

$$Z_i = A_i Z_{i-1} + B_i X_i$$
$$Y_i = C_i Z_i + D_i X_i,$$

where $Z_0 = 0$ and $A_i, B_i, C_i, D_i$ are input-dependent matrices. In the S6 block [Gu and Dao, 2024], they are parametrized as functions of $X_i$, yielding a structured recurrence.

This system admits a an attention formulation [Sieber et al., 2024, Dao and Gu, 2024]: $Y = \Phi_{\text{S6}}^X \cdot X$,

$$\Phi_{\text{S6}}^X = \begin{pmatrix} C_0 B_0 + D_0 & 0 & \cdots & 0 \\ C_1 A_1 B_0 & C_1 B_1 + D_1 & \cdots & 0 \\ \vdots & \ddots & \ddots & \vdots \\ C_N \prod_{k=1}^{N} A_k B_0 & \cdots & C_N A_N B_{N-1} & C_N B_N + D_N \end{pmatrix}. \tag{1}$$

Later options such as Mamba2 [Dao and Gu, 2024] and DeltaNet [Yang et al., 2025] also share this view, yet their parameter-efficient formulation introduces further state expansion and parameter sharing options for efficient heads.

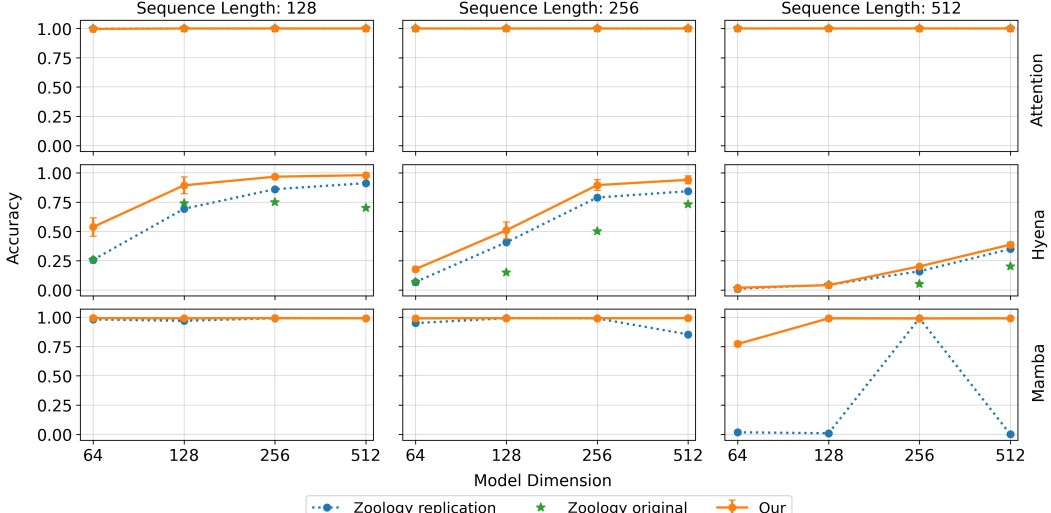

Figure 2: *Performance of 2-layers models. Results for H3 [Fu et al., 2023], RWKV [Peng et al., 2023] and Based [Arora et al., 2024] included in App. A.3. We report the official results[2] (green stars) and the replication running the original code of [Arora et al., 2023] (dotted blu line). While for replication, we used the learning rates grid by Arora et al. [2023], we note here that, due to high sensitivity to the learning rate (Fig, 1), tuning drastically affects performance. In solid orange, we provide results with a finer grid (cf. Fig.1). Careful tuning of the learning rate gives a general improvement in the performance of recurrent models. This becomes especially crucial in Mamba, where the task becomes solvable at high sequence lengths >> hidden size. The results show the mean and relative max-min errors for 3 seeds. Attention always solves the task (all curves overlap).*

## 3   Closer Look into AR performance

Building on previous research, we aim to provide an in-depth analysis of the differences and similarities between attention and recurrent models through the lens of AR. Prior studies [Arora et al., 2023] have shown that transformers are inherently well-suited for solving the MQAR task, achieving perfect accuracy regardless of model dimension, sequence length or number of key-value pairs to infer. In contrast, it was argued (both theoretically and empirically) that new recurrent models [Peng et al., 2023, Nguyen et al., 2024, Gu and Dao, 2024] can only solve MQAR if the hidden dimension is roughly equal to the sequence length (see analysis by Jelassi et al. [2024] in a related setting). However, a key aspect that has been **overlooked** in some prior works is the crucial role of optimization in recurrent models —particularly, the use of an effective grid search for the choice of learning rate.

**Hypothesis from previous works.**   Recurrent models update their hidden state (which serves as a compressed representation of past information) at each time step, using the current input. Since the model only has access to its hidden state and the current input, its ability to recall previous information depends on how effectively it compresses past data into this state. With a simplified analysis assuming uniform distribution over strings, Jelassi et al. [2024] showed that to successfully copy input strings, the hidden size needed grows linearly with the sequence length. In contrast, transformers [Vaswani et al., 2017] dynamically access all previously seen inputs through the softmax attention mechanism, allowing for the explicit computation of interactions between tokens. This makes the task of recalling already seen tokens essentially a lookup table problem when two layers work simultaneously, as described in Jelassi et al. [2024], Olsson et al. [2022].

**Results.**   Compared to previous work, in our experiments, we devoted more attention to tuning the learning rates, drastically improving the reported performance for recurrent models (see Fig. 2&1). As shown in Figure 2 and extensively in Appendix A.3, a finer grid not only enhances average performance across all settings but also proves particularly crucial for the Mamba model. With a more suitable learning rate, Mamba [Gu and Dao, 2024], which was previously shown to struggle with long sequence lengths, becomes capable of solving MQAR at relatively small hidden model sizes.

---

[2]Mamba was not included in the official work but some experiments are documented in the blog post

All experimental details for this and the next experiments are in Appendix A.2. This highlights a key takeaway for MQAR: the choice of learning rate (and optimization strategy in general) can be decisive in assessing whether a recurrent model can solve the task at all. In the case of Mamba, optimization choices become a discriminative factor, emphasizing the necessity of careful hyperparameter tuning in recurrent models, and further research for improving their high sensitivity.

To further emphasize the critical role of learning rate selection in training recurrent models, we compare the performance of Attention, Hyena and Mamba using the same grid search. Figure 1 illustrates that attention-based models maintain strong performance across a relatively wide range of learning rates. In contrast, Hyena and Mamba exhibit a different behavior: performance remains near zero for most learning rates but suddenly reaches near-optimal levels at specific values which may not be included in the grid by Arora et al. [2023]. These findings highlight a key distinction between attention-based and recurrent models: a sparse learning rate grid search can disproportionately impact their training outcomes. **This discrepancy can lead to misleading conclusions** about the capabilities of these models, emphasizing the need for careful tuning.

# 4 Effects of width/depth scaling into AR performance

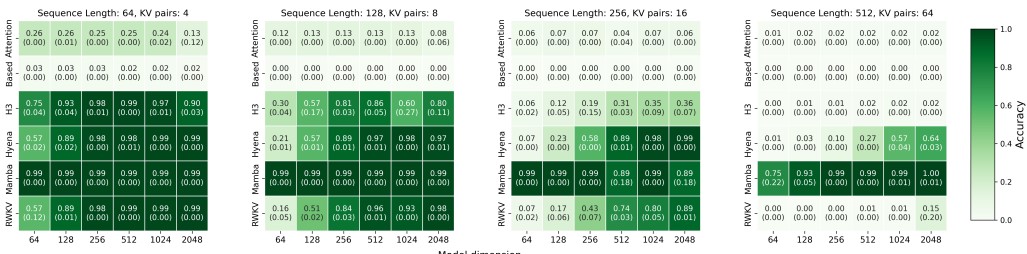

Figure 3: *Performance of 1-layer attention-based (Attention, Based) and recurrent-based (H3, Hyena, Mamba, RWKV) models on AR. We show how for recurrent models, scaling the width boosts performances. On the contrary, attention models cannot solve the task anymore as in the 2-layer setting, and performances are unaffected by the scaling in width. The results show the mean and relative max-min errors after 3 runs with different seeds.*

While our findings in Sec. 3 show that some recurrent models can exhibit improved performance on MQAR with proper learning rate tuning, we confirm that a sizeable gap with attention can still be observed for some recurrent models at low widths (e.g. Hyena vs. Attention). The experiments of Sec. 3 focused on comparisons of 2-layer architectures, at different sequence lengths and model widths. This choice stems from prior research [Olsson et al., 2022], where transformers have shown peculiar in-context learning capabilities related to the formation of induction head circuits in 2-layer models. With the intention of going beyond the setup that is known to show strengths for softmax attention, our objective in this section is to explore the effects of scaling in different configurations.

To achieve this goal, we conducted experiments analogous to Section 3 using single-layer architectures[3]. By doing so, we aim to decouple the effects of inter-communication between layers and to isolate the impact of each model's fundamental structure (attention versus recurrence) on MQAR. Beyond this, our motivation also comes from the notable connections that have been drawn between attention and recurrent models [Dao and Gu, 2024, Ali et al., 2024, Sieber et al., 2024] and on the capabilities of transformers [Sanford et al., 2024] – all of which concern 1-layer models. Our results, presented in Figure 3, reveal two key insights:

1. First, for a fixed sequence length, recurrent models always benefit from scaling in width – as was happening in 2 layers (Sec. 3). That is, expanding the hidden state dimension enhances their performance. This result aligns well with current literature [Jelassi et al., 2024, Orvieto et al., 2024]: as already mentioned, at each time step recurrent models store compressed inputs into a hidden state, which serves as a condensed representation of all past information. A larger hidden dimension facilitates less aggressive compression, allowing the model to retain more information.

---

[3]By single layer in attention and recurrent models we mean a sequence mixer followed by an MLP.

2. Attention models exhibit a surprisingly different behavior: when constrained to a single layer, they fail to solve the task and increasing the hidden dimension does not affect their performance. This is in stark contrast to their strong results in 2-layer architectures, where even the smallest model was sufficient to solve the task in the hardest setting. Interestingly, in this setting transformers are capable on average of recalling one key-value pair in every setting, suggesting a memory size issue when only one layer is present as also suggested in previous work [Sanford et al., 2024].

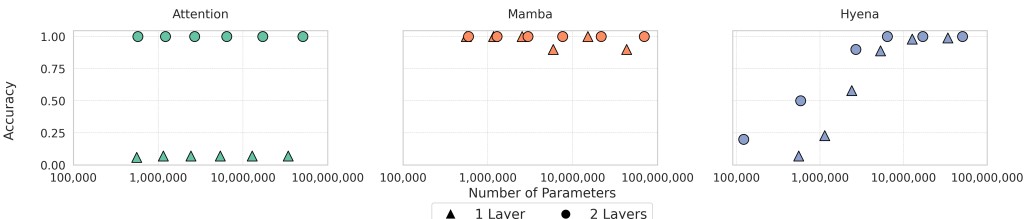

Figure 4: *Scaling models in width and depth (Seq len: 256, KV pairs: 64). Symbols with the same shape and color represent models of increasing size in the following order: 64, 128, 256, 512, 1024, and 2048. We show how rather than the number of parameters, is the way these models are scaled that impacts performance. Specifically, recurrent models benefits from scaling in width, while attention benefits from scaling in depth.*

Our findings highlight a key takeaway from our study: attention and recurrent models exhibit opposite scaling behaviors in width and depth. In other words, as shown in Fig. 4, rather than the number of parameters, it is the way these models are scaled that has most impact on their performance.

## 5  1-layer Training Dynamics and Induction Heads phenomenon

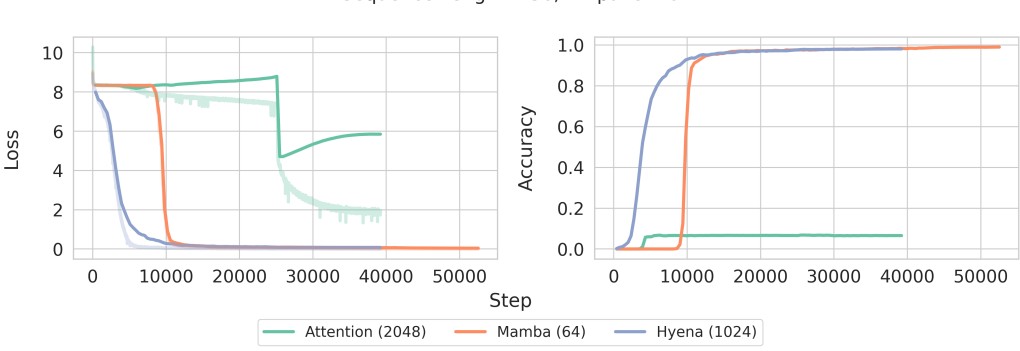

Figure 5: *Training (lower opacity) and Validation dynamics of 1-layer models. We reported within brackets the smallest width that solves the task, if possible; or otherwise the biggest width we tried (for attention). Differently from Mamba, Hyena requires the model dimension to exceed the sequence length. Both exhibit smooth learning dynamics, leading to perfect performance. Attention shows a loss bump, but without accuracy gains, suggesting an attempt to form induction heads that the single-layer transformer fails to leverage effectively.*

Sec. 4 sparked our curiosity, leading us to explore the single-layer architecture setup further – to understand why attention hits a performance ceiling while recurrent models can solve the task. This analysis is especially intriguing given the strong connections that have been proposed between attentions and Mamba in Ali et al. [2024], Dao and Gu [2024].

In this section, we analyze the training dynamics of well-tuned Hyena, Attention and Mamba models. As illustrated in Fig. 5 we identify two main patterns. First, Hyena (and similarly other non-selective recurrent models like H3 and RWKV) exhibits consistently smooth learning dynamics, with a gradual and steady improvement that eventually lead to convergence at the solution. Specifically, loss reductions align closely with increases in accuracy. Differently, attention accuracy remains largely

Table 1: *Performance of ablated 1-layer architectures. After making different modifications to the Mamba and Transformer, we report if the model is capable of solving the MQAR task. Here solving the task means achieving accuracy ≥ 95 percent with any combination of sequence length 64, 128, 256, 512 and model dimension 64, 128, 256, 512, 1024, 2048*

| Model | Solves MQAR |
|---|---|
| Attention | × |
| Attention + Conv on QKV | ✓ |
| Attention + Conv on K | ✓ |
| Attention + Conv on V | ✓ |
| Attention + Conv on Q | × |
| Mamba | ✓ |
| Mamba w\o conv1d | ✓ |
| Mamba w\o gating | ✓ |
| S6 + MLP (Mamba as a Transformer) | ✓ |

unchanged throughout training. A similar trend appears in the test loss, which remains relatively stable until a sudden bump occurs, after which the test loss settles again. This bump resembles the formation of an induction head circuit [Olsson et al., 2022], and to the best of our knowledge has previously only been observed during the training of multi-layer transformer architectures. However, as opposed to what can observe in 2-layer models, this phase transition in the loss does not correspond to an accuracy improvement for attention. Based on previous work [Olsson et al., 2022], we hypothesize that during this phase transition, the attention mechanism *attempts* to form induction heads. However, in the single-layer setting, the model lacks the expressivity needed to effectively leverage this mechanism for task resolution. Interestingly, the dynamics of Mamba is mixed:

1. Like single-layer attention models, we report a significant loss bump, reinforcing the connection between Mamba and attention mechanisms, as suggested in Ali et al. [2024], Dao and Gu [2024].
2. However, unlike transformers, Mamba can successfully solve the task even in a single-layer setting – provided the learning rate is properly tuned, similarly to other recurrent models.

Our results highlight a crucial distinction: while attention and recurrent models share some common ground, yet distinct inductive biases. Moreover, their performance is in strong interaction with the optimization algorithm at hand (in our case, Adam [Kingma, 2014]), as we also saw in Figure 1. Understanding these nuances is key to optimally leveraging both architectures, perhaps also towards hybrid models [Waleffe et al., 2024, Dao and Gu, 2024].

# 6 Are SSMs and Transformers really similar?

Our results so far highlight key differences between Transformers and SSMs, particularly Mamba, in the context of AR. Notably, while Mamba demonstrates greater expressivity—successfully solving the task even in a single-layer setting—it presents optimization challenges in terms of learning rate stability. In contrast, Transformers exhibit remarkable stability across a wide range of suitable learning rates during training in the 2-layer setting. We highlight that each of these layers includes an MLP block processing channels.

To address this discrepancy, we conduct a series of ablation studies aimed at:

1. Modifying the architectures of both models to better align them (architectures are in appendix A.1) and identify the source of Mamba's superior performance, summarized in Table 1 and
2. Exploring new architectural variants that promote more stable training dynamics.

**Convolutions.** Inspired by [Li et al., 2024], we begin by adapting the attention mechanism to resemble Mamba. We incorporate a 1D convolution before the Query,Key and Value matrix projections to brings in locality, enabling the model to solve MQAR with just one layer. Interestingly, we observe that applying the convolution to either the Key or Value matrix alone is sufficient to achieve the same performance gains. These observations suggest that the 1D convolution may be

a central factor behind Mamba's effectiveness. However, we find that even after removing the convolution, the model retains the ability to solve the task.

**Backbone ablation.** We further modify the Mamba architecture to make it closer to the Transformer architecture by: (1) removing the gating mechanism, and (2) replacing the standard Mamba block with a sequence mixer (as in S6), followed by an MLP—mirroring the Transformer's architecture. Despite these alterations, Mamba continues to perform well when properly tuned, suggesting a notable degree of robustness in the design of its fundamental block.

**Newer architectures.** To better understand what contributes to training stability, we also evaluate architectural variants designed for improving the Mamba architecture and solve the MQAR task. In particular, we test Mamba2 [Dao and Gu, 2024] and DeltaNet [Yang et al., 2024b] as shown in Figure 6. While performance of Mamba2 is slightly more stable, Transformer-level robustness is only achieved by DeltaNet. A closer look at the DeltaNet update rule reveals that its mixing is based on Householder matrices. As such, the off-diagonal terms such as $C_N \prod_{k=1}^{N} A_k B_0$ do not necessarily incur in vanishing gradients. Instead, in both Mamba and Mamba2, $A_k$ includes a decay rate that induces vanishing gradients and fast decay of off-diagonal terms, as recently pointed out by Trockman et al. [2024]. We hypothesize this is the main distinction unlocking stable optimization in DeltaNet.

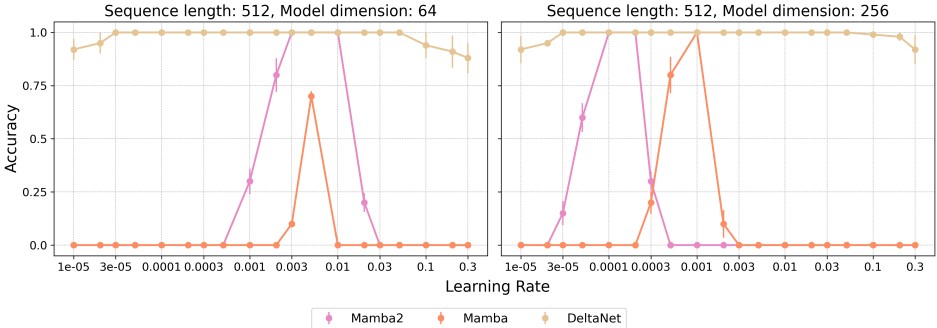

Figure 6: *We show the performance of Mamba, Mamba2 and DeltaNet in the 1-layer setting using the same learning rate grid search. Here we show how having a bigger hidden state marginally helps stability, as in Mamba2 and especially in DeltaNet. Results could be show to a maximum of model dimension of 256 because DeltaNet implementation doesn't support greather dimensions. The results show the mean and relative max-min errors after 3 runs with different seeds.*

# 7 Discussion and Conclusions

In this work, we used MQAR as a benchmark to compare attention and recurrent models at a small scale. Our findings shed additional light on how the underlying mechanisms of these models influence their performance. Specifically, we showed that recurrent models are highly sensitive to optimization, with their performance significantly affected by the choice of learning rate. This underscores the need for further research to improve their stability. Additionally, we observed contrasting scaling behaviors: recurrent models benefit from the increased width and hidden state size, whereas transformers struggle with MQAR in a single-layer configuration. Interestingly, despite their poor performance, single-layer transformers exhibit training dynamics resembling the induction head phenomenon, previously reported only in multi-layer settings. Instead, Mamba displays similar behavior but successfully solves the task. Finally, through the ablations study, we showed how the performance of Mamba is robust to specific architectural components such as gating and convolution, and how other similar architectures can enhance performance and stability. Our findings suggest overlaps between the optimization landscapes of Mamba and Attention, yet with crucial differences related to expressivity, to study further. Looking ahead, we think that exploring other synthetic reasoning tasks and architectural changes could provide further insights into the mechanisms behind these models. Evaluating these areas is an important direction to refine our understanding of modern sequence models.

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

# A  Appendix

## A.1  Modified architectures

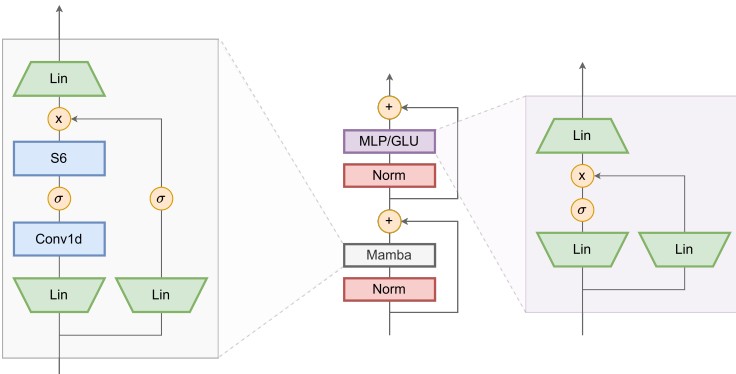

Figure 7: *Architecture of Mamba following the architecture of a Transformer given by sequence mixers interleaved by MLPs*

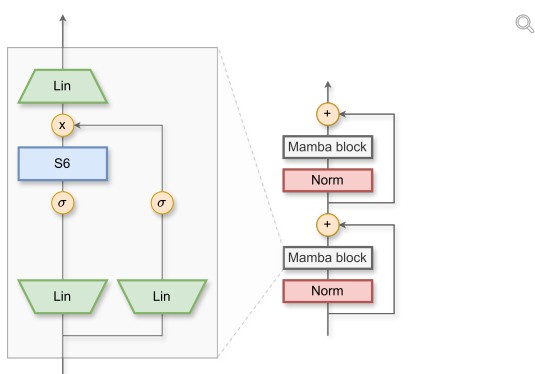

Figure 8: *Architecture of Mamba without the conv1d*

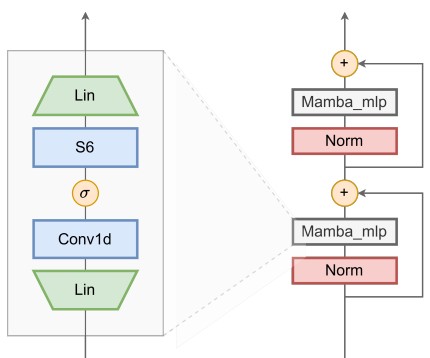

Figure 9: *Architecture of Mamba without the gate*

## A.2 Experimental details

In this section, we describe the experimental setup used throughout our study. Clearly outlining these details is crucial for interpreting the results presented in subsequent sections. Our implementation is inspired by methodologies from Zoology [Arora et al., 2023].

**Data.** The dataset consists of sequences of tokens representing key-value pairs. Tokens are sampled from a fixed vocabulary of $8,192$ tokens. Within each sequence, key-value tokens are assigned randomly, ensuring that the model cannot learn a static mapping. Consequently, each sample is independent, requiring the model to infer the role of tokens in context rather than relying on memorization. The synthetic dataset is structured with four specific sequence lengths, each paired with a corresponding number of key-value pairs to recall:

- 64 tokens with 4 key-value pairs;
- 128 tokens with 8 key-value pairs;
- 256 tokens with 16 key-value pairs;
- 512 tokens with 64 key-value pairs.

For the first three sequence lengths, the ratio of key-value pairs to sequence length is $1 : 16$, whereas for the longest sequence, the ratio is $1 : 8$, making it the most challenging case. For each sequence length, a dedicated dataset is created, consisting of $100,000$ training samples and $3,000$ test samples. Model evaluation is performed by training each model on a specific sequence length and subsequently assessing its performance on that same length.

**Models** Our experiments utilize a total of six main models + others used in the ablation studies:

- Two attention-based models: Attention and Based.
- Four recurrent models: H3, Hyena, RWKV and Mamba.
- Other Ablations such as Attention + Convolution, Mamba without specific components etc.

Each model is tested across six model dimensions: 64, 128, 256, 512, 1024 and 2048. Additionally, models are implemented in two configurations: 1-layer and 2-layer. Notably, a "layer" in our context refers to the concatenation of two blocks: a sequence mixer (e.g., attention, RWKV, etc.) followed by an MLP. Thus, a 1-layer model consists of two blocks, aligning with the terminology used in prior work [Arora et al., 2023, Olsson et al., 2022]. Positional information is used only in attention and Based.

**Training and Evaluation** We used GPU A100 with 80GB of memory in all our experiments. We trained for 50 epochs using AdamW as optimizer, weight decay 0.1, warmup duration 10%, linear warmup. All the experiments took between 10 minutes and 18 hours based on the model architecture, the model dimension and the sequence length. The batch size varied depending on the sequence length: 128 for sequence length 512, 256 for sequence length 256 and 512 otherwise. Each configuration (combining model type, model dimension, and sequence length) undergoes a learning rate sweep to identify the optimal learning rate. The reported accuracy for each configuration corresponds to the best performance achieved across the tested learning rates. We want to highlight that the accuracy reported should be interpreted as the average percentage of key-value pairs correctly labeled. Specifically, achieving $50\%$ accuracy with sequence length $64$ and $4$ as relative number of key-value pairs means that on average the model recalls correctly 2 values given 4 keys. To ensure robustness, all experiments are conducted using three random seeds (42, 123, and 777), with results reported as the mean and standard deviation across these trials.

## A.3 Full Tables

---

[4]Mamba was not included in the official work but some experiments, with different settings compared to ours, are documented in the blog post.

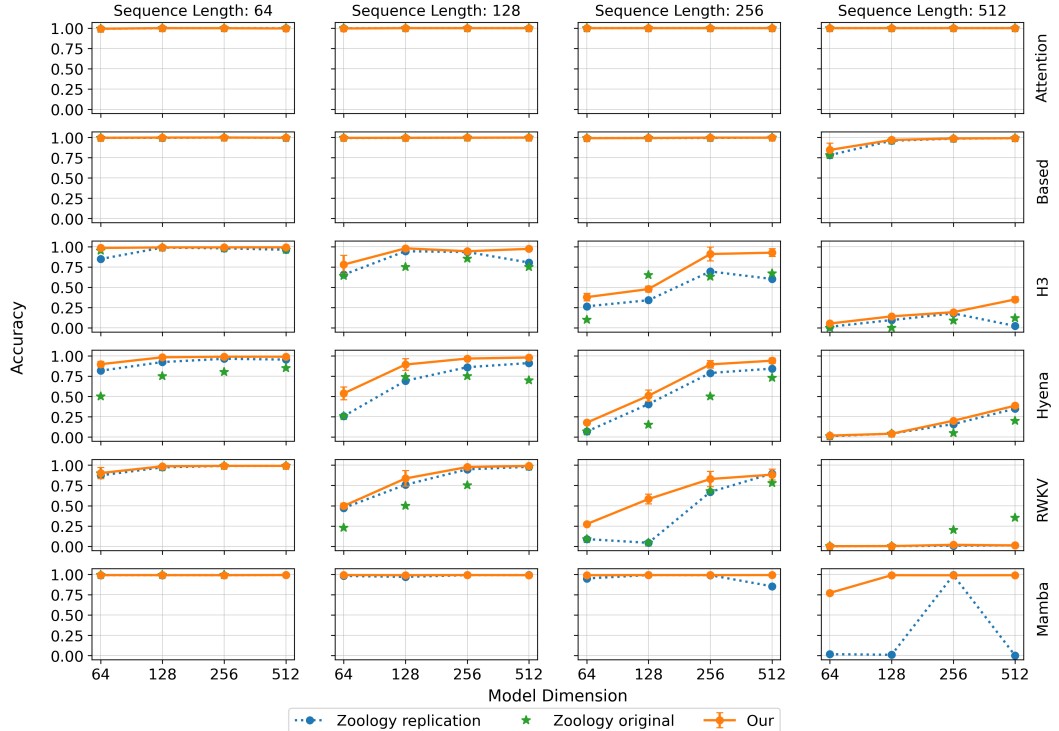

Figure 10: *Performance of 2-layers models. We report the official results[4] (green stars) and the replication running the original code of [Arora et al., 2023] (dotted blu line). While for replication, we used the learning rates grid by Arora et al. [2023], we note here that, due to high sensitivity to the learning rate (Fig, 1), tuning drastically affects performance. In solid orange, we provide results with a finer grid (cf. Fig.1). Careful tuning of the learning rate gives a general improvement in the performance of recurrent models. This becomes especially crucial in Mamba, where the task becomes solvable at high sequence lengths >> hidden size. The results show the mean and relative max-min errors for 3 seeds. Attention always solves the task (all curves overlap).*

