# OpenReview forum: "When recalling in-context, Transformers are not SSMs"
_NeurIPS.cc/2025/Conference — Submitted to NeurIPS 2025_

### Official Review · Reviewer_obvz · 2025-06-20

**Clarity:** 4
**Significance:** 4
**Originality:** 3
**Rating:** 5
**Confidence:** 4

**Summary:**

1. This paper systematically investigates the differences between Transformer and SSM (State Space Model) architectures in the context of associative recall (AR) tasks.

2. Several observations are particularly noteworthy:
   (1) SSMs exhibit extreme sensitivity to the learning rate, with good performance only achievable within a very narrow range.
   (2) The preferred scaling strategies differ: Transformers benefit more from increasing depth, whereas SSMs tend to perform better when scaled in width.
   (3) Architectural components contribute differently to the performance of Transformers and SSMs. For instance, the combination of attention and convolution (Attn+Conv) is effective in addressing the MQAR task.

**Questions:**

1. Can the authors provide a mechanistic or theoretical explanation for why Mamba and Hyena have such narrow “optimal” learning-rate windows, in contrast to Transformers?
Even a heuristic or empirical visualization (e.g., gradient norms over time) would greatly strengthen the optimization narrative.
2. How do your findings (e.g., SSMs' optimization sensitivity) generalize to real-world downstream tasks?

**Ethical Concerns:**

["NO or VERY MINOR ethics concerns only"]

**Final Justification:**

After carefully considering the rebuttal, author clarifications, and discussions among reviewers, I remain confident in my positive assessment. The paper addresses a relevant problem, presents a sound and well-executed approach, and demonstrates solid empirical results. In my view, these strengths clearly outweigh the concerns raised, and the work meets the standard for acceptance.

**Limitations:**

While the authors suggest in the NeurIPS checklist that limitations are discussed in Section 7, they are not adequately addressed in the main text.
## Suggestion:
Please include a paragraph in the conclusion that reflects on:
- The generalizability of MQAR-based findings to real-world tasks (e.g., language modeling, reasoning).
- The absence of theoretical grounding for observed optimization behavior.

**Paper Formatting Concerns:**

The text in multiple figures is too small to read comfortably. Please increase the font size in these figures.

**Quality:**

3

**Strengths And Weaknesses:**

# Strengths
## Quality
- systematic comparison of Transformers and SSMs on MQAR;
- comprehensive ablation (convs, gating, backbone swaps) strengthens causal claims.
## Clarity
- well written, structure is clear;
- figures cleanly support each claim;
- paper follows a clear motivation → method → result chain.
## Significance
- clear identification of optimization sensitivity. shows that SSMs' poor AR performance in prior work was often due to suboptimal learning rate.
- highlights opposite scaling laws (depth vs. width), guiding future hybrid designs.
## Originality
- Introduces a fine-grained optimization analysis seldom seen in prior SSM work.

# Weaknesses
## Quality
- results confined to synthetic MQAR; no real-world downstream tasks
## Originality
- no new model; contributions are analytical rather than methodological.

---

> ### Author Rebuttal · Authors · 2025-07-29
>
> We would like to express our gratitude for the comprehensive feedback provided. The reviewer's direct assessment is greatly appreciated, and we will address each of the raised concerns and suggestions in our response below:
>
> ---
>
> **Concerns about generalization on downstream tasks**
> We thank the reviewer for highlighting concerns regarding the generalizability of our findings. The fact that our results are confined to the synthetic MQAR task, without direct validation on real-world downstream applications, is a point we have also carefully considered.
>
> Our decision to focus on the MQAR task is intentional and grounded in recent literature identifying in-context recall as a primary bottleneck for SSMs. Our choice is motivated by findings like those in "An Empirical Study of Mamba-based Language Models" (Waleffe et al., 2024), which states: "Our results show that while pure SSMs match or exceed Transformers on many tasks, they lag behind Transformers on tasks which require strong copying or in-context learning abilities." This positions our study not on an arbitrary task, but on a capability that is fundamental to improving SSMs.
> Furthermore, our methodology aligns with a recent line of research using targeted tasks to understand core capabilities predictive of large-scale performance (e.g., in "Unlocking State-Tracking in Linear RNNs Through Negative Eigenvalues" by Grazzi et al. 2025). Within this paradigm, MQAR is the prototypical benchmark for in-context recall. The link to downstream performance is quantitatively established by  "Zoology: Measuring and Improving Recall in Efficient Language Models" by Arora et al. 2023, demonstrating that failings on associative recall tasks account for up to 80% of the performance gap between modern recurrent models and Transformers on downstream language benchmarks. This suggests that solving MQAR is not just an isolated challenge but a critical stepping stone towards closing the overall performance gap.
>
> Therefore, while we did not run experiments on downstream tasks ourselves, we posit that our findings, particularly those concerning the optimization sensitivities of SSMs, have direct implications for real-world applications. Investigating the conditions necessary to unlock the capabilities of models is critical for real-world performance. We believe this provides a principled basis for expecting our conclusions to generalize, and we will clarify this perspective in the camera-ready version.
>
> ---
>
> **Explanation on the learning dynamics of RNNs**
> We thank the reviewer for raising questions on the theoretical understandings of our work. The request for an explanation for the observed optimization sensitivity of Mamba and Hyena is highly relevant, and we agree that a deeper investigation strengthens our claims. We would like to clarify that the primary contribution of our work is indeed analytical: to empirically identify fundamental confounders, such as the interplay between optimization and expressivity of models, that arise when training modern SSMs. This focus aligns with foundational questions in the recent literature, as explored in works like the blogpost "*On the tradeoffs of SSMs and transformers*" by Gu 2025, for which MQAR serves as an effective and targeted benchmark.
> While a complete theoretical derivation is outside the scope of our current work, we can offer two well-grounded intuitions that explain the narrow "optimal" learning rate windows we observed:
>
> - First, our results empirically position SSMs closer to classical RNNs. This connection is critical because the optimization challenges of the latest are extensively documented. The literature has long established that RNNs suffer from notoriously difficult learning dynamics, including treacherous loss landscapes with sharp minima, extensive plateaus, and numerous saddle points. This makes them extremely sensitive to hyperparameter choices, especially the learning rate. This phenomenon is detailed in seminal works such as "*On the difficulty of training recurrent neural networks*" by Pascanu et al. 2013, "*Learning Recurrent Neural Networks with Hessian-Free Optimization*" by Martens et al. 2011 and illustrated visually in papers like "*Recurrent Neural Network: vanishing and exploding gradients are not the end of the story*" by Zucchet et al. 2024 (e.g., Figure 2a). Our findings suggest that SSMs, as a form of recurrent model, inherit these challenging optimization characteristics, which stand in contrast to the more stable attention mechanism of Transformers.
>
> - Second, building on this connection, we hypothesize that this inherent sensitivity is amplified by the specific architectural design of SSMs like Mamba. The use of a structured, often diagonal, state transition matrix A is the key to their computational efficiency and ability to model long-range dependencies. However, we posit that this constrained structure creates a trade-off. Unlike the dense dot-product matrices in attention, this diagonal parameterization may lead to a more brittle optimization landscape, where slight deviations from the optimal learning rate can more easily lead to unstable or non-convergent training dynamics. We also empirically verified this by looking at the gradient norms at initialization, which are often close to zero.
>
> As the reviewer kindly suggested, we will update the paper to make these connections and hypotheses more explicit.
>
> ---
>
> **Limitations**
> As suggested by the reviewer, we added a new specific section where we more specifically discuss the limitations of our work. In particular, we focused on two main aspects: the lack of experiments in more complex language tasks to see if our findings hold, and the lack of theoretical proof of what causes the learning instability of linear models.
>
> ---
>
> **Readability of text in figures**
> We thank the reviewer for pointing out the difficulty in reading text in figures (we assume, in particular, the text in figure 3). We already changed the relative labels to make it more readable.

---

> > ### Comment · Reviewer_obvz · 2025-08-05
> >
> > I'm grateful to the authors for pointing out that the extreme sensitivity of SSMs' architecture to the learning rate is caused by its own architectural design, which is highly inspiring. Designing architectures with stable learning rates will also be of great significance in the future.

---

> > > ### Author Response · Authors · 2025-08-06
> > > **Response to Reviewer**
> > >
> > > We thank the reviewer for the positive and insightful feedback. Your comment that our findings are "highly inspiring" and significant for future work on designing architectures is greatly valued.
> > >
> > > Spurred by the review process, we conducted a new set of experiments to further strengthen our claims and demonstrate their generalizability beyond the initial MQAR setting. We tested our hypotheses on the well-known copy task, a benchmark previously used to highlight performance gaps between Transformers and SSMs (e.g. "*Repeat after me: Transformers are better than State Space Models at copying*", Jelassi et al. 2024). In summary, our new results show that:
> > > - SSMs exhibit a very narrow optimal learning rate window also on the copy task, which may have been missed in prior work, leading to conclusions of their underperformance.
> > > - Our core thesis that SSMs benefit from width while Transformers benefit from depth holds true. We demonstrate that making SSMs deeper simply to match a Transformer's parameter count creates a suboptimal architecture that is bound to fail.
> > >
> > > These findings reinforce our paper's central argument that the perceived limitations of SSMs can often be attributed to specific optimization and architectural scaling choices rather than fundamental design flaws. We believe these new experiments substantially strengthen the impact and robustness of our work. We have provided a more detailed summary of these results, including performance tables, in our response to reviewer Zbu9.
> > >
> > > We hope that these new findings address any remaining concerns and would be grateful if you would consider them in your final evaluation of our paper. We thank the reviewer again for their time and constructive engagement.

---

### Official Review · Reviewer_2wet · 2025-06-27

**Clarity:** 3
**Significance:** 2
**Originality:** 2
**Rating:** 4
**Confidence:** 3

**Summary:**

The paper empirically investigates differences between Transformer and SSM‑based architectures (notably Mamba and Hyena) on the Multi‑Query Associative Recall (MQAR) task. Through extensive learning‑rate sweeps, width vs. depth scaling experiments, and ablation studies, the authors show that (1) SSM variants are far more sensitive to learning‑rate choices than Transformers; (2) Transformers solve MQAR robustly with sufficient depth, whereas SSMs require extreme width scaling; and (3) single‑layer training dynamics (e.g. “induction‑head”‑like loss bumps) manifest differently in each family. These observations are demonstrated on synthetic benchmarks and detailed in a series of figures and tables.

**Questions:**

- Are the compared models (e.g., Mamba, Hyena, Transformer variants) linear or non-linear? Given that prior equivalence results (e.g., Katharopoulos et al., 2020; Dao et al., 2022) assume linearity, clarifying this is crucial to assess whether your findings contradict existing theory or simply fall outside its scope.

- Can the authors clarify what aspect of the observed non-equivalence is surprising or novel, given that general non-linear SSMs and Transformers are not expected to behave the same?
- The authors show that SSMs (e.g., Mamba) are far more sensitive to learning rate than Transformers, but it’s unclear *why* this is the case. Is this due to gradient vanishing, parameter coupling, ill-conditioned kernels, or some other architectural factor? A deeper analysis—either empirical (e.g., gradient norms, Jacobian spectrum) or theoretical—would significantly strengthen the main claim.
- The authors point out that different models exhibit different sensitivity to learning rate, with Transformers showing more stable optimization. However, could you explain more concretely why this leads to the conclusion that *“Transformers are not SSMs mainly because of their optimization dynamics”*? What exactly makes this difference in optimization behavior a basis for architectural non-equivalence?

**Ethical Concerns:**

["NO or VERY MINOR ethics concerns only"]

**Final Justification:**

To me, it is obvious that Transformers are different from SSMs. This paper makes some detailed comparisons on this point. The authors’ response did not change my view on the matter, so I am keeping my original score.

**Limitations:**

yes

**Paper Formatting Concerns:**

Null

**Quality:**

2

**Strengths And Weaknesses:**

**Strengths**

- The authors perform dense learning‑rate grid searches (Fig. 1) and systematic width/depth sweeps (Figs. 2–4), revealing subtle biases in optimization and scaling.
- Single‑layer training curves (Fig. 5) and ablations (Fig. 6, Table 1) are well‑presented and reproducible, offering insight into why SSMs struggle without careful tuning.
- This paper reveals that prior underperformance of SSMs may be owed more to hyperparameter choices than fundamental limitations, which is valuable for practitioners.

**Weakness**

- The paper’s main contribution is essentially a counterexample showing that SSMs and Transformers behave differently under certain settings (e.g., on MQAR). While this is clearly demonstrated, the finding itself is not particularly surprising. The scope of the paper is limited to a single synthetic benchmark (MQAR), making it hard to assess whether the findings generalize to real-world tasks.
- Prior works such as **Katharopoulos et al. (2020)** only claim equivalence between Transformers and SSMs under specific assumptions, such as **linearity**. This paper does not clearly state whether it studies linear or non-linear models, but given the use of standard architectures like **Mamba** and **Hyena**, it is likely working in the **non-linear** regime, where such equivalence obviously no longer holds. Because of that, the observed non-equivalence between Transformers and SSMs feels somewhat expected, rather than a novel or surprising discovery.

---

> ### Author Rebuttal · Authors · 2025-07-28
>
> We thank the reviewer for the time and effort invested in their detailed and helpful review. We appreciate the balanced perspective offered, and below we provide clarifications to the concerns and questions that were raised to improve our work.
>
> We believe, and this we sensed as also a general feeling among the other reviewers, that our claim is novel in the literature and non-trivial. We also, however, agree with you: SSMs are mathematically different from transformers. Yet the non-equivalence proposed in the title is not to be taken literally, as a mathematical fact, but in strict regard to capabilities. Quoting Reviewer Zbu9, "*The comparison between Transformers and SSMs is a crucial and active area of research in the machine learning community*".
>
> ---
>
> **Transformers and SSMs are clearly different, hence results seem trivial**
>
> We will update the discussion so that our title and claims are not interpreted as trivial, given the apparent mathematical differences between Transformers and SSMs. The "*are*" in the title refers to practical capabilities concerns driven by recent works studying similarities between these models, rather than equivalences. We thank the reviewer for this opportunity to improve clarity. This also gives us the opportunity to be more precise in our claims and statements.
>
> - We are only discussing post-SSM (post S4) architectures. Understanding how such architectures differ fundamentally from (1) plain linear attention and (2) softmax attention is a very open, vibrant, and crucial nontrivial topic. Just a few weeks ago, Albert Gu posted a blog on this on his website, with the title "On the Tradeoffs of SSMs and Transformers".
>
> - The reviewer is correct that prior work, such as *Katharopoulos et al. 2020*, established equivalence between Transformers and RNNs *only under strict linearity assumptions*. However, note that SSMs are more complex, deep, and nonlinear objects — and several modern works are revisiting the claim you cite much more carefully: take, for instance, "*The Hidden Attention of Mamba Models*" by Ali et al. 2024. In particular, check their equation (17), which shows how Mamba can recover exponential parametrization (such as the one in softmax attention). Further, we point to "*Understanding the differences in Foundation Models: Attention, State Space Models, and Recurrent Neural Networks*" by Sieber et al. 2024 in the last NeurIPS, comparing normalization strategies between Mamba and Attention: as in softmax denominator in Attention vs. $\Delta$ parameter in Mamba. Note that these works deviate significantly from the linear setting in *Katharopoulos et al. 2020*.
>
> - Our analysis takes inspiration from the recent (approximate) claim in the popular Mamba2 paper "*Transformers are SSMs: Generalized Models and Efficient Algorithms Through Structured State Space Duality*" by Dao et al. 2024. This paper claims that the Transformer's attention mechanism itself can be seen as the sequence mixer of an SSM. Our title is a clear response to this, but we are not arguing here about mathematical details. Our interest is in capabilities.
>
> - To directly answer the reviewer's question, the models we analyze (e.g., Mamba, Hyena, and RWKV) are indeed non-linear in their full architectural form. In these models, the non-linearity typically arises from recurrent LSTM-like gating mechanisms and MLP blocks that follow the sequence mixer. The core theoretical equivalence we investigate, as proposed in *Dao et al. 2024*, concerns the sequence mixing block itself (i.e. only input-dependent gating and normalization). Our goal was to test whether the proposed similarities at a sequence-mixing level translate to equivalent capabilities in practice.
>
> What we found is that SSMs can be as expressive as Transformers, but they are more challenging to optimize. As such, while they can have similar capabilities (a.k.a, quoting again the title, "*Transformers are SSMs*"), they behave differently in practice.
>
> We therefore strongly advise against interpreting our title in isolation and will revise our discussion to ensure that our intended message and context are clearly conveyed. We thank the reviewer for bringing up this issue.
>
> ---
>
> **Choice of the task**
> We agree that understanding generalization to broader language tasks is a crucial direction to have a more complete and convincing work. As stated also in other responses, this choice stems from both practical reasons (comprehensive thorough ablations) and insights in the literature. Citing "*An Empirical Study of Mamba-based Language Models*" by Waleffe et al. 2024: "Our results show that while pure SSMs match or exceed Transformers on many tasks, they lag behind Transformers on tasks which require strong copying or in-context learning abilities (e.g., 5-shot MMLU, Phonebook) or long-context reasoning".
>
> This sentence clearly points to studying in-context recall as a fundamental direction towards improving SSMs. In essence, this is not an arbitrary task, but represents what SSMs struggle with the most. We offer here a pointer to optimization as the issue, rather than expressivity.
>
> Our choice of restricted focus further aligns with the recent line of research that uses targeted synthetic tasks to isolate and understand specific capabilities predictive of large-scale language model performance. For example, similar focused approaches have been used to analyze state tracking (e.g., in "*Unlocking State-Tracking in Linear RNNs Through Negative Eigenvalues*" by Grazzi et al. 2025). Within this research paradigm, MQAR is widely considered the prototypical benchmark for assessing in-context critical capabilities. As shown in "*Zoology: Measuring and Improving Recall in Efficient Language Models*" by Arora et al. 2023, performance on recall tasks like MQAR explains over 80% of the performance gap between Transformers and certain linear recurrent models on downstream tasks.
>
> ---
>
> **Analysis on the challenge of optimization**
> We thank the reviewer for recognizing how our work focuses on various experiments that expose how linear models are hard to optimize. We agree that a definitive theoretical analysis would significantly strengthen our claims, leading to a higher quality final work, which we ultimately seek.
>
> Our current work's primary focus was to empirically surface and analyze key confounders (e.g. "*The trade-offs between optimization and capability"*) that arise when comparing SSMs and Transformers, a topic of active discussion in recent literature (e.g., in the blogpost "*On the tradeoffs of SSMs and transformers*"  by Gu 2025). We used the MQAR task as a controlled benchmark to expose these differences. Regarding why SSMs like Mamba exhibit far greater sensitivity to the learning rate, our experiments and the existing literature point to two complementary hypotheses:
>
> - Firstly, we hypothesize that the sensitivity is intrinsically linked to the architectural design of modern SSMs. The diagonal structure of the state matrix (A matrix in the SSM formulation) is a cornerstone of how these models efficiently handle long-range dependencies. However, our results suggest this design creates a trade-off, gaining long-context capability at the cost of a more brittle and sensitive optimization landscape compared to the softmax attention mechanism of transformers.
>
> - Secondly, our work empirically demonstrates that the learning dynamics of SSMs on the MQAR task are remarkably similar to those of classical RNNs. This connection is critical because the difficulty of training RNNs is a well-documented phenomenon. A rich body of literature has explored their challenging loss landscapes, characterized by plateaus and sharp saddle points that make optimization notoriously difficult (e.g., "*On the difficulty of training recurrent neural networks*" by Pascanu et al. 2013, "*Recurrent Neural Network: vanishing and exploding gradients are not the end of the story*" by Zucchet et al. 2024 (Figure 2a) and "*Learning Recurrent Neural Networks with Hessian-Free Optimization*" by Martens et al. 2011). By showing that SSMs inherit these dynamics, we place their optimization challenges within this well-established context. We also note that while we experimented with other optimization techniques like learning rate scheduling, the choice of the base learning rate consistently proved to be the dominant factor, further underscoring this core architectural sensitivity.
>
> This leads to the reviewer's other excellent key question: why does this difference in optimization behavior justify the conclusion that "Transformers are not SSMs"? We argue that an architecture's identity is defined not only by its forward pass expressivity but also by its learnability. If two models, claimed to be dual at their core, require fundamentally different conditions to be successfully optimized for a key task, they cannot be considered functionally equivalent in practice.
> The optimization process is not an external implementation detail; it is an intrinsic property that determines whether an architecture's theoretical capabilities can be realized. The Transformer's success is partly owed to its relatively stable and forgiving optimization dynamics, which allowed it to scale in a way previous recurrent models could not. Our finding that SSMs revert to RNN's optimization challenges is therefore a fundamental difference. It is this divergence in the practical ability to learn a solution that forms the basis of our claim: their profound difference in optimization dynamics is a core architectural distinction, not a superficial one.

---

> > ### Comment · Reviewer_2wet · 2025-08-05
> >
> > Thank you for all your responses and explanations. I have decided to maintain my original rating.

---

> > > ### Author Response · Authors · 2025-08-06
> > > **Response to Reviewer**
> > >
> > > We thank the reviewer again for their time dedicated to improving our work.

---

### Official Review · Reviewer_kRPJ · 2025-07-02

**Clarity:** 4
**Significance:** 3
**Originality:** 3
**Rating:** 4
**Confidence:** 4

**Summary:**

The authors study the performance of SSM and transformer models on the associative recall task. They note that SSMs are especially sensitive to learning rate, which provides a new interpretation of previous results. They then deepen their investigation into, e.g. one-layer performance for Transformers and SSMs as well as into modified Mamba and Transformer architectures.

**Questions:**

## Possible typo:
On line 290, the authors say "replacing the standard Mamba block with a sequence mixer (as in S6)". This is a little confusing, because the  standard mamba with selection mechanism is the S6 Mamba. In the original paper that introduced Mamba, an S4-Mamba was considered only in some extra experiments. I assume this is a typo or some misunderstanding on my part. It would be good to clarify this.

## Title:
I find that the current title is memorable, but a bit confusing. I.e. of course Transformers are not SSMs, not when recalling in context, and not in any other situation. I understand what the authors mean, but I think this version misses the opportunity to hint at actual specific results/problems.

The first key insight introduced in the paper is learning rate sensitivity, and the key question is about expressive power vs training difficulties. I.e. "are SSMs weak or just finicky?" Maybe it'd be good to hint at that in the title, e.g. "SSMs are expressive but sensitive", or even "SSM models are underestimated," "Don't underestimate SSMs" etc.

The title didn't affect my evaluation, and I don't insist that the authors change it. But I believe it would benefit the paper to consider some alternatives.

**Ethical Concerns:**

["NO or VERY MINOR ethics concerns only"]

**Final Justification:**

I've read the authors rebuttal and other reviews. There is a fairly general consensus and I retain my score for the same reasons I listed in my last comment.
On a separate note, I wish the authors best of luck in finding the right title for the paper, as multiple reviewers independently noted the slight issues with the current one. This issue, however, did not affect my score.

**Limitations:**

The paper does not discuss its limitations in sufficient depth. The most natural question to address is whether or not the findings would generalize to practical applications. And whether under optimal training regimes SSMs would actually be able to compete with transformers in practical applications. I believe that if the authors re-focus the paper and more clearly state what the main contribution is, it'd naturally make it easier to address the limitations as well. I.e. if the authors take the risk on a slightly stronger main claim, they can then address the alternative explanations etc. in the limitations section.

I have no concerns regarding the potential negative societal impact.

**Paper Formatting Concerns:**

No concerns

**Quality:**

3

**Strengths And Weaknesses:**

## Strengths
- The question considered is extremely relevant.
If the limitations of SSM architectures are less substantial than previously considered, it'd be very impactful.
- The paper is well-written and is a pleasure to read.
- The experiments are reasonable and well-executed.
- The authors do a good job placing their work in the context of existing research.

## Weaknesses
- Some fundamental questions are not addressed.

The authors mention the key theoretical limitation of SSM models--their memory capacity has to be proportional to the size of its hidden state. But there is little discussion of whether and how the new results affect this.

I, for example, was expecting the authors to argue in the discussion that while this limitation remains valid, perhaps in practice, SSM potential is much higher than previously thought. Instead, this topic never came up in the discussion. I believe it's crucial to devote more attention to it. This result is the most impactful part of the paper.

- Partial lack of focus.
Again, the main contribution of the paper seems to be "SSM models are unjustly underestimated because they are very sensitive to the learning rate." Unfortunately, it is not introduced as the main focus of the paper. Instead, it's mentioned as the first of many other results.
I believe that this creates an unfocused impression.
Instead of diving deeper into a detailed investigations on when exactly learning rate matters, does it differ based on the optimizer, etc., the authors greatly broaden the scope of the paper, going as far as to consider alternative architectures for both Transformer and Mamba blocks.
Instead of one highly impactful and thoroughly tested result, they present a wide array of results, most of which (especially in the new architectures section) do not (in my opinion) have enough depth to be relied upon in practice or further research.

To illustrate my point, consider the last few sentences from the paper:
"Finally, through the ablations study, we showed how the performance of Mamba is robust to specific architectural components such as gating and convolution, and how other similar architectures can enhance performance and stability. Our findings suggest overlaps between the optimization landscapes of Mamba and Attention, yet with crucial differences related to expressivity, to study further."

Unfortunately, this is a very weak "final point" to make. Essentially, it states that a) the Mamba architecture can be tweaked a little without losing performance and b) "there are similarities but also differences between SSMs and Transformers."
Both of these conclusions are almost self-evident. Regarding part a) - the Mamba architecture was not introduced as a result of extensive architecture search, so it's not surprising that not every component of it is absolutely crucial. Part b) is true for almost any two models.

## Conclusion
Overall, the main result is still very interesting and potentially highly impactful. So despite the limitations, I believe that at present, it's a (very borderline) accept.

To make the paper stronger, the authors should
1) Strengthen the experimental support for learning rate sensitivity (specifically with different optimizers).
2) Streamline the framing of the paper (clearly state which results are central and which are secondary).

I will, of course, read the authors' rebuttal and other reviews. I will adjust my score up or down if new considerations come to light that I haven't yet accounted for.

---

> ### Author Rebuttal · Authors · 2025-07-28
>
> We sincerely thank the reviewer for the thoughtful and constructive evaluation. We found the detailed feedback on our work's contributions, limitations, and potential improvements to be very valuable.
>
> In the following, we address the points and questions raised.
>
> ---
>
> **Improving Focus**
> We thank the reviewer for this highly constructive feedback. The reviewer has highlighted two related issues: a partial lack of focus and a failure to connect our results to fundamental theoretical questions. The comments have provided us with the clarity to improve substantially our flow, and we will restructure accordingly with more precise claims.
>
> - Let us start by assessing the practical relevance of the task and issue we study. Even though SSMs are considered successful (largely due to their inference speedup), *slower training* of Mamba-based models has been reported in e.g., "*An Empirical Study of Mamba-based Language Models*" by Waleffe et al. 2024. Quoting their abstract, "*Our results show that while pure SSMs match or exceed Transformers on many tasks, they lag behind Transformers on tasks which require strong copying or in-context learning abilities (e.g., 5-shot MMLU, Phonebook) or long-context reasoning. In contrast, we find that the 8B Mamba-2-Hybrid exceeds the 8B Transformer on all 12 standard tasks we evaluated[...].*". We believe that non-hybrid SSMs have yet to show their potential, possibly due to the optimization issues we presented. This discussion will be included in our revision.
>
> - Let us now present our central claim: SSM models can show suboptimal performance on recall benchmarks not due to fundamental limits on their expressive power (i.e., limited memory as opposed to KV cache), but because they suffer from optimization issues — slowing down or directly blocking training.
>
> - Our experiments are logical steps in supporting this thesis, and we will restructure our paper backbone to follow a clearer path: To start, our choice of the MQAR task allows us to ask — in a controlled setting where we can perform many ablations — if SSMs can theoretically match Transformers on a task highly correlated with recall performance (as shown in "*Zoology: Measuring and Improving Recall in Efficient Language Models*" by Arora et al. 2023) where theoretically Transformers were shown to be superior. This strategy is inspired by other works that focus on one specific capability to predict models' performance at scale (e.g. state tracking in "*Unlocking State-Tracking in Linear RNNs Through Negative Eigenvalues*" by Grazzi et al. 2025).
>
> - Our initial experiments lead directly to the central finding: SSMs can succeed, but only within a smaller learning rate window compared to transformers. This discovery of extreme optimization sensitivity will be presented as the core phenomenon under investigation, not just one result among many. This finding also gives hope to future researchers towards constructing improved SSMs: we are likely not suffering from a bottleneck in expressivity.
>
> - The reviewer suggested delving deeper into learning rate sensitivity by using other optimizers. We note that Adam is among the very few methods capable of optimizing modern architectures consisting of heterogeneous blocks, see e.g. claims on the Hessian in "*Why Transformers Need Adam: A Hessian Perspective*" by Zhang et al 2024. This central topic in the optimization community was also developed in the RNN world by Zucchet et al. 2024 in "*Recurrent neural networks: vanishing and exploding gradients are not the end of the story*": they show that the different decay factors in recurrent linear memory are tackled efficiently by the Adam optimizer but not by simpler methods such as SGD — a problem which dates back to early investigations on RNN landscapes such as in "*Learning Recurrent Neural Networks with Hessian-Free Optimization*" by Martens et al. 2011. A future avenue might include, e.g. an investigation of Muon, which is however hard to set up and experimental at this stage. For completeness and to also assess our previous claim we have run additional experiments with an SGD optimizer. We noticed that SGD fails consistently among all the models analyzed (transformer included), as can be expected from recent literature. High complexity methods such as second-order optimizers are unfortunately not an option, since they would be costly for language modeling.
>
> - The final section, which the reviewer saw as an unfocused broadening of scope, is stemming from (a) curiosity in regards to the induction heads literature (recall mechanisms in transformers are constructed with 2 transformer layers) and (b) claims regarding necessity of gating for fundamental language modeling tasks (e.g. "*Hungry Hungry Hippos: Towards Language Modeling with State Space Models*" by Fu et al. 2023). This will be reframed as a targeted ablation study to probe which elements in the architecture are affecting results — crucially, note that the Mamba and the Transformers backbone are different! By comparing how architectural tweaks affect SSMs versus Transformers, we are not just cataloguing results, but providing evidence that the optimization challenges are likely *rooted in the core recurrent mechanism of SSMs*. With this, our purpose is to further eliminate confounders: issues are not due to *backbone differences* or contingent to depth, but rooted in token mixing. While this might be obvious to some, we found it necessary to be thorough.
>
> We thank the reviewer for the incentive to revisit our claims. We will adapt the paper accordingly and are open to further suggestions.
>
> ---
>
> **Week conclusions**
> We thank the reviewer for the valuable feedback, and agree that the paper's conclusion needs a revision. We agree that being confident in this part plays a crucial role in the quality of the work. We will align our discussion with our central claim and advocate for future research on this topic, as we are confident that solving or improving optimization issues will substantially improve SSM training. This is also why we think NeurIPS exposure would be extremely beneficial.
>
> Regarding our side claims (that we will not emphasize in the conclusion): there is a flip side to the story, which warns us that 1-layer architectural comparisons bringing SSMs and attention closer (e.g., the one in "*The Hidden Attention of Mamba Models*" by Ali et al. 2024) are also prone to dangerous simplifications and wrong conclusions regarding the attention mechanism (that is, the *flip side* of our argument). We demonstrated that while certain modifications (e.g., convolutions) were vital for the Transformer to succeed in our one-layer setting, the core SSM mechanism was surprisingly resilient to the removal of its gating and convolution components. This comparative result points to a fundamental difference in how these architectures derive their power for the associative recall task. It also shows that transformers with one layer (as studied in works that compare such architectures mathematically) are not as powerful as SSMs, and the Mamba backbone does not entirely drive this distinction (Mamba without convolution performs similarly).
>
> ---
>
> **Title**
> We thank the reviewer for defining our title as "*memorable*" and for their constructive suggestions. We agree that a title should be as clear and indicative of the paper's contents as possible, and we are certainly open to considering alternatives for the final version.
>
> The current title was chosen deliberately to be in direct dialogue with recent work, "*Transformers are SSMs: Generalized Models and Efficient Algorithms Through Structured State Space Duality*" by Dao et al. 2024. That paper made the compelling theoretical claim that the sequence mixing mechanism in Transformers can be viewed through the lens of an SSM Model. Intrigued by this, our research aimed to investigate whether this equivalence holds up under the lens of a critical practical capability: in-context recall. Our experiments, however, revealed a stark divergence in behavior. We found that when tasked with associative recall, SSMs exhibit optimization challenges, particularly an extreme sensitivity to the learning rate, that make them behave much more like classical RNNs than like Transformers. Therefore, our title was intended as a direct counterpoint to the theoretical claim of the aforementioned work, highlighting that the similarity is not entirely accurate and can lead to wrong conclusions.
>
> The reviewer is entirely correct that our title, in making this pointed claim, may miss the opportunity to advertise our specific findings, such as the trade-off between expressive power and optimization sensitivity. We hope that this explains the reason why we chose the current title. However, since we seek to produce a high-quality work, we are open to finding a new version that preserves the important intellectual context of our work while better communicating our key contributions to a broader audience.
>
> ---
>
> **Limitations**
> We completely agree with the reviewer and will use the additional space in a potential camera-ready submission to develop on our limitations: clearly, our work poses a question which we do not answer: "how to alleviate optimization issues in modern RNNs?". We believe our thorough study of recall is a valid way to kickstart this research, which is motivated by results in *Arora et al. 2023* and by the empirical results in *Waleffe et al. 2024*.
>
> Another limitation is that, to keep focus and provide detailed ablations, our study is limited to (many settings in) the MQAR benchmark. We believe this benchmark is extremely relevant, but it would be of interest to also study this phenomenon on other problems, specifically when evaluating future solutions.
>
> ---
>
> **Typo**
> Thanks for spotting this. The corrected sentence would be "replacing the standard Mamba block with the individual sequence mixer S6, followed by an MLP"

---

> ### Comment · Reviewer_kRPJ · 2025-08-08
> **Rebuttal acknowledgement**
>
> I thank the authors for their responses to my and other reviews. At present, I will keep my score, though in my mind it is now a "high borderline" close to "clear accept" paper. Part of the reason I can't increase the score right away is that when the concerns are about framing & focus, it's hard to judge whether they will be successfully addressed, even if the intention is there.
>
> That said, I really appreciate that the authors understand where I'm coming from and are open to some restructuring of their paper which, I am certain, will make it stronger.
>
> P.S. re: the title. Thank you for clarification. I now better see the motivation behind the original title, but I still feel that it can be improved to highlight the key insights. The "Transformers are SSMs" title is strong because it says something quite surprising (in an exaggerated manner). "Transformers are not SSMs" does not have the same punch, in my opinion. Especially when it's preceded by "when recalling in-context."
>
> Just as a point to consider: I'm actually familiar with the "Transformers are SSMs" paper, but the reference in the title did not "click" "for me. When tying the title to another work like that, the only people who will "get it" are those who read the other paper recently enough to instantly remember it. Especially since the title is not structured the same. So the target audience becomes a subset of the other paper's audience. I agree that "Transformers are SSMs" work is very impactful, but it's not "Attention is all you need" level of impactful, so it's not clear whether anchoring yourself to that work is wise.
>
> If the authors feel very strongly about "Transformers are not SSMs", I'd suggest "Transformers are not SSMs, after all: *key conclusion.*". This way, the structure of the title will make it clear that it's a reference to another work. But I personally vote for something completely different.
>
> At the end of the day, it is, of course, the authors' decision.

---

> > ### Author Response · Authors · 2025-08-08
> > **Response to reviewer kRPj**
> >
> > We sincerely thank the reviewer again for the detailed and constructive feedback.
> >
> > We are particularly grateful for the suggestions on improving the paper's focus and framing. We believe that the proposed changes will significantly enhance the clarity and impact of our work, which is indeed our objective. We also appreciate the comments on the title that is currently more directed to a niche public and will carefully consider alternatives for the final public version to better appeal to a broader audience.
> >
> > Thank you for your open-mindedness and for indicating your willingness to reconsider your score, and we hope our response has clarified our direction.

---

### Official Review · Reviewer_Zbu9 · 2025-07-02

**Clarity:** 4
**Significance:** 3
**Originality:** 3
**Rating:** 4
**Confidence:** 4

**Summary:**

This paper provides a timely and insightful critical evaluation of Transformers and modern recurrent models, specifically State-Space Models (SSMs) like Mamba, focusing on their performance in associative recall (AR) and multi-query associative recall (MQAR) tasks. The authors present compelling arguments and experimental evidence challenging some previously held beliefs about the limitations of recurrent architectures. Specifically, recent theoretical works have claimed that in terms of representations, SSMs are transformers, however, these works typically ignore the effect training dynamics have on the representation. The authors provides empirical evidence showing the need to take into consideration optimization dynamics in claims of representation equivalence. The authors also show that modifications to Mamba - Mamba2 and DeltaNet helps alleviate the performance penalty and this may be due to better gradients in the modified models. Further, the experiments also show that SSMs benefit from width increase, while transformers benefit from depth increase although the claim is validated only for depths 1 and 2.

These are the claims made in the paper

Claim 1: Unlike standard Transformers, the performance of recurrent models is highly sensitive to the choice of learning rate, which can determine whether they solve a task. Overlooking this factor may lead to incorrect conclusions about their capabilities.

Claim 2: Recurrent models benefit significantly from increased width, whereas single-layer attention models notably fail to solve AR and are unaffected by width scaling. Conversely, attention-based models exhibit strong performance when scaled in depth (e.g., 2-layer architectures).

Claim 3: The paper observes that single-layer Transformers exhibit training dynamics reminiscent of induction heads (a phenomenon previously exclusive to deeper models), but without corresponding accuracy improvements. Mamba, in contrast, shows smoother training dynamics and a steep performance increase akin to induction heads even in a single-layer setup.

Claim 4: Through ablation studies, the authors suggest that Mamba's performance advantages are robust even when its convolutional or gating components are removed or modified. They also propose architectural modifications (e.g., 1D convolution) that enable single-layer Transformers to solve MQAR.

All claims are substantiated through experiments in the MQAR dataset.

**Questions:**

1. I have noticed that papers that show the better performance of transformers compared to SSMs, when they are representationally similar uses positional embeddings for transformers but not state space models. This is the same protocol that is used in this paper also. Have you tried adding positional embeddings to RNNs? It makes sense to compare against this too because the main claim to disprove in the paper is "ssms are transformers" and it is important to remove this source of variation in the experiment.
2. Does the claims in the MQAR task translate to issues in training transformers vs SSMs in large language tasks?
3. It is possible that since RNNs have the ability to compress information, it may perform better in a task requiring only fuzzy outputs, compared to exact copy required for MQAR tasks.
4. Does the relationship between depth/width increases in transformers and RNNs translate to depths more than 2?

**Ethical Concerns:**

["NO or VERY MINOR ethics concerns only"]

**Final Justification:**

I raised a number of issues related to the toy nature of the problem and the scope of the claims made, The title, in my opinion, is very broad given the experimental results provided in the paper. The title as the authors note are inspired from "Transformers are SSMs" which previously showed that Linear Transformers can be mathematically identical to an SSM. From this it is very clear that ordinary transformers are not SSMs, so it is not clear to me in what way the authors wanted to convey transformers were not SSMs. There is no theoretical exposition or deeper insights that can be obtained from the results, but it looks like the empirical evaluation is sound. In light of this, I retain my original score.

**Limitations:**

The limitations of the work is not discussed as claimed in the paper.

**Paper Formatting Concerns:**

Typo

- Fig 6. DeltaNet implementation doesn’t support "greather" dimensions
- Fig 6. Grammar. Results could be show to a maximum of model dimension of 256... this is grammatically incorrect.

**Quality:**

3

**Strengths And Weaknesses:**

## Strengths

- The comparison between Transformers and SSMs is a crucial and active area of research in the machine learning community. This paper makes a contribution by dissecting their operational differences in an experimental setting in a toy dataset.
- The figures provide compelling evidence for the claims raised in the paper. The work also suggests that performance reported in earlier works on SSMs may have suffered from the sensitivity issue.
- The analysis of training dynamics and the discovery of induction-head-like phenomena in single-layer attention models are thought-provoking and could spur further research into the learning mechanisms of these architectures. To my understanding the work that showed the presence of attention heads in 2-layer transformer models also showed that these heads are not present in 1-layer transformers. So, this evidence contradicts the earlier claim, although the author admits that the metric indicating the presence of attention heads does not translate to actual performance.

## Weaknesses

- While the title is attention-grabbing, the paper's conclusion that "transformers are not SSMs mainly because of their optimization dynamics" could benefit from a more precise articulation of the interplay between optimization, architectural expressivity, and fundamental differences. There could be more of a theoretical exposition in this direction especially because experiments are conducted in a toy dataset.
- The study is primarily confined to (MQ)AR tasks. While these are excellent for studying in-context learning, generalizing the findings to broader applications like large-scale language modeling or other downstream tasks would require further empirical validation.
- The statistical power is poor, with only 3 seeds performed to validate the experiments. Given the short timeframe required to run the experiments and the available large computation capabilities, it is unclear why significance tests are not conducted.
- A deeper theoretical investigation into _why_ recurrent models exhibit such a narrow optimal learning rate window, beyond empirical observation, would significantly strengthen the key claims.
- The code to reproduce the results are not provided. The authors suggest looking at another codebase which the work is based on but the experiments conducted are different.

Overall, the paper is well-written and the claims made in the paper are substantiated in the experiments. The work is timely for the discussion about the relation between recurrent state space models and transformers. However, the experimental evaluation is performed in a toy dataset and the statistical power is poor (only 3 seeds). The insights from these experiments may still fall into the same trap of prior works discussed in the paper. There is also no theoretical exposition to strengthen the claims of the paper and as a result the question "why there is a disconnect" is not fully substantiated. It could be that more sophisticated optimization procedures currently employed in SSMs - like cosine scheduling can get over the learning rate sensitivity.

---

> ### Author Rebuttal · Authors · 2025-07-28
>
> We want to firstly thank the reviewer for the rich and insightful review, highlighting strengths, weaknesses and suggestions to improve our work. It is our ultimate goal to produce a solid and interesting paper to drive future research towards better SSMs.
>
> We address all your points below.
>
> ---
>
> **Choice of the title**
> We understand the concern of the reviewer on the strong claim in our title, and we are open to discussing other options; however, we would like to give more context on our decision. The title choice draws inspiration from the work "*Transformers are SSMs: Generalized Models and Efficient Algorithms Through Structured State Space Duality*" by Dao et al. 2024, where the authors demonstrate that the sequence mixing strategy of the attention mechanism in transformers parallels that of an SSM. Intrigued by this, in our work, we test the in-context learning abilities of such models with the MQAR task. Our experiments show how SSMs behave more like RNNs in terms of optimization, and drastically differently from softmax attention, as opposed to what can be deduced from the aforementioned work.
>
> ---
>
> **Is MQAR enough?**
> The reviewer correctly noticed that our analysis revolves only around the synthetic task of associative recall, suggesting that further work should be performed on other tasks and perhaps also pure language modeling.
>
> Let us start, as we did for reviewer kRPj, by quoting the abstract by "*An Empirical Study of Mamba-based Language Models*" by Waleffe et al. 2024 :"*Our results show that while pure SSMs match or exceed Transformers on many tasks, they lag behind Transformers on tasks which require strong copying or in-context learning abilities (e.g., 5-shot MMLU, Phonebook) or long-context reasoning*". This sentence clearly points to studying in-context recall as a fundamental direction towards improving SSMs.
>
> More in general, our purpose in this paper is to bring to light an optimization-related *failure mode* of RNNs on a task associated with fundamental LM capabilities. Our results show how pure expressivity arguments (as the ones presented in the literature we cite) are not sufficient to fully understand performance. Our work can be seen as a *counterexample* on a task of practical interest, simple enough to perform many ablations, and is similar in spirit to many recent papers addressing a specific yet fundamental capability (e.g., copying, state tracking, length generalization, etc). While it is for sure interesting to show if other tasks also showcase similar issues, we believe results on this problem alone are sufficient to kickstart and motivate research around improving SSMs. To summarize, our objective is to show that severe optimization issues "can happen" when dealing with tasks associated with fundamental capabilities. MQAR gave us the opportunity to empirically study and report this issue, and its interplay with model size and task difficulty.
>
> Further, we like to emphasize once more that our choice aligns well with the recent trend of focusing on one specific capability (e.g. state tracking in "*Unlocking State-Tracking in Linear RNNs Through Negative Eigenvalues*" by Grazzi et al. 2025) to predict language model performances at scale. MQAR is the prototypical task for assessing the ability to recall tokens seen in context. More specifically, "*Zoology: Measuring and Improving Recall in Efficient Language Models*" by Arora et al. 2023 shows how this capability is responsible for 80% of the gap in performance between transformers and some linear models.
>
> ---
>
> **Statistical relevance of the results**
> We understand the reviewer's concern about the statistical relevance of our experiments. Since for us it is fundamental to have a solid and convincing analysis, we decided to run our experiments with three seeds to ensure a minimal level of statistical relevance — despite the large number of experimental settings and hyperparameter sweeps in our work.  We believe that our findings, such as poor learning rate stability, are interesting even when limited to 3 seeds, as failing to optimize on 3 seeds out of 3 for most learning rates has practical relevance.
>
> We, however, believe the reviewer is correct in that adding more seeds increases the statistical validity of our findings: we will add two more seeds in our experiments for a potential camera-ready version, and thank the reviewer for the suggestion.
>
> ---
>
> **Theoretical explanation of RNNs optimization**
> We agree with the reviewer that a deeper theoretical investigation, though highly difficult due to the high nonconvex nature of the problem, would strengthen our claims.
>
> The main point we delivered is that expressivity studies on MQAR alone may not be sufficient for designing or understanding linear RNNs, as optimization issues must also be considered. Actually, we *already know* from the literature that training RNNs is hard — but the reviewer is correct in that a thorough discussion of the literature is currently lacking. We will fix this in our updated version.
>
> One classical paper devoting attention to this topic is "*On the difficulty of training recurrent neural networks*" by Pascanu et al. 2013, exploring the loss landscape of nonlinear recurrences, in particular vanishing and exploding gradients resulting from optimization of models that are iterated over time. On the linear recurrent side, a recent paper "*Recurrent Neural Network: vanishing and exploding gradients are not the end of the story*" by Zucchet et al. 2024 showed that RNN linearity does not *per se* imply easier optimization, and that long-range memory alone poses challenges also in SSMs: see their Figure 2a, showing a hard-to-escape saddle point as memory increases. We also point out, and will in our revised version, that many works have already tried to solve this optimization issues in the pre-SSM era, such as "*Learning Recurrent Neural Networks with Hessian-Free Optimization*" by Martens et al. 2011. We believe adding a paragraph discussing these results would improve the value of our work, especially in regards to future research. We thank the reviewer for making us work towards a more comprehensive discussion.
>
> ---
>
> **Code Publishing**
> Since we wanted to ensure absolute consistency with the original MQAR setup, all our experiments are based on the original work of *Arora et al. 2023*. We will happily publish the code related to our experiments for the camera-ready version.
>
> ---
>
> **New section for limitations**
> We will comment on the fact that insights from other tasks will be interesting for future research, and that work is needed to understand if optimization can be unlocked with better architecture/initialization.
>
> ---
>
> **Use of positional embeddings**
> We thank the reviewer for the interesting question. While adding positional embeddings in RNNs is not generally believed to add significant value, since the positional information is already given implicitly by the recurrence at each time step, we also thought the question was worth investigating. We will add the results below in our revision.
>
> Below we show results for PE (absolute, relative, learned…) on model dimension 256, sequence length 512, and 2 layers.
>
> | Type of PE | Mamba | Hyena | RWKV | H3 |
> | :-------- | :----------: | :---------: | :---------: | :----: |
> | No PE | 0.99 $\pm$ 0.00 |  0.30 $\pm$ 0.11 |  0.19 $\pm$ 0.09 |  0.24 $\pm$ 0.15 |
> | Absolute PE |  0.99 $\pm$ 0.00 |  0.33 $\pm$ 0.11 |  0.22 $\pm$ 0.09 |  0.25 $\pm$ 0.10 |
> | Relative PE | 0.99 $\pm$ 0.00 |  0.31 $\pm$ 0.04 |  0.22 $\pm$ 0.09 |  0.27 $\pm$ 0.07 |
> | Learned PE |  0.99 $\pm$ 0.00 |  0.29 $\pm$ 0.08 |  0.24 $\pm$ 0.13 |  0.28 $\pm$ 0.11 |
>
> Results do not seem much affected by PE, yet most importantly, we did not notice any change in learning rate sensitivities.
>
> ---
>
> **RNNs might be better at Fuzzy recall**
> As the reviewer correctly notices, new RNNs have indeed shown remarkable capabilities in recalling fuzzy representations of input in context, given their inherent ability to compress information. We believe our failure mode analysis (i.e. poor learning rate sensitivity) is most interesting in the noiseless setting, given the outstanding performance of attention-based models. The noisy setting will likely introduce other issues, yet perhaps only for attention, confounding optimization with expressivity issues.
>
> ---
>
> **What happens in the case of layer-depth > 2**
> We thank the reviewer for the interesting question. We were also intrigued by this in the early stage of this work, and we are happy to share, also in our updated manuscript, the results we obtained —after very careful tuning— with the following table:
>
> | N layers | Attention | Based | Mamba | Hyena | RWKV | H3 |
> | :- | :-: | :-: | :-: | :-: | :-: | :-: |
> | 4 layer | 0.99 $\pm$ 0.00 |  0.99 $\pm$ 0.00 |  0.99 $\pm$ 0.00 |  0.31 $\pm$ 0.09 |  0.19 $\pm$ 0.12 |  0.27 $\pm$ 0.13 |
> | 3 layer | 0.99 $\pm$ 0.00 |  0.99 $\pm$ 0.00 |  0.99 $\pm$ 0.00 |  0.29 $\pm$ 0.07 |  0.20 $\pm$ 0.10 |  0.23 $\pm$ 0.11 |
> | 2 layer | 0.99 $\pm$ 0.00 |  0.99 $\pm$ 0.00 |  0.99 $\pm$ 0.00 |  0.30 $\pm$ 0.11 |  0.19 $\pm$ 0.09 |  0.24 $\pm$ 0.15 |
> | 1 layer | 0.02 $\pm$ 0.00 |  0.01 $\pm$ 0.00 |  0.99 $\pm$ 0.00 |  0.28 $\pm$ 0.13 |  0.17 $\pm$ 0.11 |  0.19 $\pm$ 0.11 |
>
> The analysis spans architectures with 1 to 4 layers, models of size 256 and sequence length of 512 (we chose this as a sufficiently challenging setting for reference, but the same applies to other settings).
>
> As shown in the Table, the major differences arise in the 1 and 2 layer settings, whereas the results with 3 and 4 layers are very close to those in the 2 layer setting, with both architectures. This aligns with works on Induction Heads, where the recall mechanism is studied and already identified in 2-layer models.
>
> ---
>
> **Typos**
> We would like to thank the reviewer. Those have now been corrected.

---

> > ### Comment · Reviewer_Zbu9 · 2025-08-01
> > **Response to the Rebuttal**
> >
> > I thank the authors for the response, since my main weaknesses on the limited empirical evaluation and theoreical analysis still stands, I keep my original score. The authors agree that these are indeed weaknesses of their current work. The authors also agree that similar problem has been raised before and there have been works addressing them, so merely reiterating the result is not a sufficient contribution.

---

> > > ### Author Response · Authors · 2025-08-05
> > > **Response to reviewer Zbu9**
> > >
> > > We thank the reviewer for their continued engagement and for clarifying their remaining concerns. We understand the main criticism is that our results are limited to the empirical setting of MQAR only.Though we believe MQAR properly reflects some of the limitations and modern questions around SSMs (see quote above by *Waleffe et al., 2024*), we agree that trying out other tasks is scientifically interesting.
> > >
> > > This feedback spurred us to conduct further experiments to provide another perspective on the optimization dynamics of SSMs that generalizes beyond our initial task. To this end, we have expanded our analysis to another fundamental benchmark: the copy task.
> > > This task is particularly relevant as prior work (e.g., "*Repeat after me: Transformers are better than State Space Models at copying*" Jelassi et al., 2024) has also used it to show performance gaps between Transformers and SSMs. In particular, *Jelassi et al. 2024* affirm that, due to their finite hidden size, SSMs solve the task only if the number of training examples increases exponentially compared to the examples needed for a transformer:
> > >
> > > | # Training Examples | 10^4 |  10^5 | 10^6| 10^7 |
> > > | :-: | :-: | :-: | :-: | :-:|
> > > | Transformer | 20% | 100% | 100% | 100% |
> > > | Mamba | 0% | 0% | 10%| 100% |
> > >
> > > Our central thesis is that such conclusions can be inadvertently skewed if the specific optimization issues are not taken into account. Specifically, our new experiments on the copy task were designed to test this hypothesis directly (with a fixed number of training examples) with two key major claim of our work:
> > > - The narrow optimal learning rate window for SSMs compared to Transformers as shown in the table below. We hypothesized that previous studies might not have performed a sufficiently granular sweep, thereby missing the optimal configuration for SSMs.
> > >
> > > | Learning Rate | 5e-2 |  3e-2 | 1e-2| 5e-3 |  3e-3 | 1e-3|5e-4 |  3e-4 | 1e-4|5e-4 |  3e-4 | 1e-4|
> > > | :-: | :-: | :-: | :-: | :-:| :-: | :-: | :-: | :-: | :-:| :-: | :-: | :-: |
> > > | Transformer | 0% | 0% | 100%| 100% |100% | 100% | 100%| 100% |100% | 100% | 100%| 100% |
> > > | Mamba | 0% | 0% | 0%| 0% |0% | 0% | 0%| 0% |100% | 0% | 100%| 96% |
> > >
> > > - Prior work often increases the depth of SSMs to match the parameter count of Transformers for a "fair" comparison. However, a core claim of our paper is that in our setting SSMs and RNN-like models benefit from width, while Transformers benefit from depth. We show that equating parameters by increasing SSM depth creates a suboptimal architecture that is bound to underperform.
> > >
> > > | Architecture | # Layers |  Width | # Parameters (M) | Accuracy (%)|
> > > | :-: | :-: | :-: | :-: | :-:|
> > > | Transformer | 12 | 1024 | 150| 100% |
> > > | Mamba| 12 | 1024 | 80| 0% |
> > > | Mamba| 24 | 1024 | 150| 16% |
> > > | Mamba| 12 | 1408 | 150| 100% |
> > >
> > > These findings demonstrate that the conclusions of prior work might not be due to a fundamental limitation of SSMs on this task, but rather a direct consequence of the specific optimization and scaling challenges we identify and analyze in our paper. We believe this new set of experiments significantly strengthens our contribution by showing that the principles we identified (width vs. depth scaling, learning rate sensitivity) are not an isolated artifact of the MQAR task but apply to other fundamental tasks.
> > >
> > > We will integrate these new findings into the revised manuscript. We are grateful for the reviewer's feedback, as it has pushed us to produce what we believe is now a substantially more robust and impactful paper. We hope that these added results will address the reviewer's concerns and persuade them to reconsider their evaluation of our work.

---

> > > > ### Comment · Reviewer_Zbu9 · 2025-08-06
> > > >
> > > > I thank the authors for providing new results. Copy task even though it is interesting, is still a synthetic task and the statistical power is still weak. Synthesis tasks are typically chosen because large scale experiments can be run and better statistical relevance can be obtained. If the main claim in the title is to be substantiated in a rigorous way, it needs to be clarified to in what ways they the two models are not equivalent. An existence of a negative result by itself is not interesting, the experiments and theoretical exposition should provide more context to the claims.
> > > >
> > > > The papers the authors refered to "Transformers are SSMs: Generalized Models and Efficient Algorithms Through Structured State Space Duality", "Unlocking State-Tracking in Linear RNNs Through Negative Eigenvalues" and "Repeat after me: Transformers are better than State Space Models at copying" provide theoretical grounding or better exposition of the main claims. In light of these fundamental issues, I retain my original score.

---

> > > > > ### Author Response · Authors · 2025-08-06
> > > > > **Response to Reviewer**
> > > > >
> > > > > We thank the reviewer for the detailed and thoughtful feedback. We appreciate your taking the time to review our new results. Your comments are invaluable and provide a clear direction for us to improve our work in the future.

---

### Note · Authors · 2025-08-12

We wish to extend our sincere gratitude to all the reviewers for their time and insightful feedback. This discussion helped us to refine our focus and strengthen the contributions of our work. Clarity and completeness are our absolute priority.

Here we summarize our main contributions and the modifications spurred by this dialogue:

- Our choice to focus on MQAR is grounded in literature highlighting in-context recall as a fundamental capability of LMs (e.g. Olsson et al. 2022, Arora et al. 2023, Waleffe et al. 2024).  Quoting the latter, “both Mamba and Mamba-2 models lag behind Transformer models on tasks which require strong copying or in-context learning abilities”. Our work stemmed from our curiosity in understanding this phenomenon.
- Our extensive experiments (~20k GPU hours) highlight that in-context performance of SSMs is severely affected by optimization issues. This is our main contribution, which we further develop by ablating on depth and impact of architectural components (e.g., convolutions, PE). We conclude that SSMs behave quite differently compared to Transformers on MQAR: while careful tuning of lr can deliver performance, this does not seem to be affected by the number of layers, lack of PE, or particular backbone components in Mamba. In contrast, 2-layer Transformers robustly perform.
- Our results highlight the optimization issue on a relevant task, and spurred by the reviewers' curiosity, we broadened our empirical validation to the copy task. This additional analysis confirms that our core findings—namely, the extreme lr sensitivity of SSMs and their preference for scaling width over depth—are not isolated to MQAR, potentially explaining performance gaps reported in prior work (e.g. Jelassi et al. 2024).
- We are particularly thankful for the encouragement to be more confident in our claims. Our central message is that an architecture's identity is defined not only by its theoretical expressivity but also by its learnability. Our findings provide empirical evidence that on these fundamental tasks, the optimization dynamics of SSMs resemble those of classical RNNs. This serves as an important counterpoint to recent works exploring similarities between these architectures (e.g. Dao et al., 2024, Gu 2025 blogpost).

Our revised paper, fortified by this constructive dialogue, now presents a more complete and impactful contribution to the ongoing research on understanding similarities and differences between SSMs and Transformers.

---

### Decision · Program_Chairs · 2025-09-17

**Decision:**

Reject

**Comment:**

The paper provides an empirical evaluation comparing Transformers and SSMs on (multi-query) associative recall tasks, with a focus on optimization dynamics, highlighting some limitations of recurrent architectures, as well as ways to overcome them via larger widths, or through more careful selection of the learning rate. The reviewers agreed that these were interesting findings, but they also found the analysis of these findings somewhat shallow, with little investigation into what is behind the observed differences between the two models. I encourage the authors to dig a bit deeper into some of these findings in a future submission, through further theoretical or empirical investigations, and perhaps to check their validity beyond the simple MQAR setup considered here.